# Experimental Research of Selected Lattice Structures Developed with 3D Printing Technology

**DOI:** 10.3390/ma15010378

**Published:** 2022-01-05

**Authors:** Paweł Bogusz, Arkadiusz Popławski, Michał Stankiewicz, Bartłomiej Kowalski

**Affiliations:** Faculty of Mechanical Engineering, Military University of Technology, 00-908 Warsaw, Poland; pawel.bogusz@wat.edu.pl (P.B.); michal.stankiewicz@wat.edu.pl (M.S.); bartlomiej.kowalski@student.wat.edu.pl (B.K.)

**Keywords:** 3D topologies, lattice structures, 3D printing, additive manufacturing technique, experimental research, energy absorbing test, compression curves

## Abstract

This paper presents the results of the experimental research of 3D structures developed with an SLA additive technique using Durable Resin V2. The aim of this paper is to evaluate and compare the compression curves, deformation process and energy-absorption parameters of the topologies with different characteristics. The structures were subjected to a quasi-static axial compression test. Five different topologies of lattice structures were studied and compared. In the initial stage of the research, the geometric accuracy of the printed structures was analysed through measurement of the diameter of the beam elements at several selected locations. Compression curves and the stress history at the minimum cross-section of each topology were determined. Energy absorption parameters, including absorbed energy (AE) and specific absorbed energy (SAE), were calculated from the compression curves. Based on the analysis of the photographic material, the failure mode was analysed, and the efficiency of the topologies was compared.

## 1. Introduction

Intensive developments in modern industry results in search for more and more unconventional technological solutions. The most durable and rigid mechanical structure with the lowest possible weight is often required to be engineered. To achieve the expected effect, research on cellular materials produced using additive methods was initiated; this enables the design of complex spatial structures through the development of these manufacturing technologies. Numerous review papers on lattice structures and their applications [1,2,3,4,5,6], illustrating the enormous potential of these materials have been published; moreover, the technological development of additive methods only improves the results.

These lattice structures can be divided into regular and irregular based on their cellular nature [7]. Regular cell structures are distinguished by a periodically repeating and interconnected topology of elementary cells. Regular structures are distinguished by their typical orthotropic mechanical properties, low density while maintaining high strength properties, high levels of energy absorption and a high stiffness-to-weight ratio [8,9].

Lattice structures are mostly in the form of regular structures. These are lattice models consisting of beams with dimeters of a few millimetres. The lattice structure mainly consists of repeated, identical elements with three-dimensional shapes, arranged in a regular pattern [10,11].

Some of the most commonly studied strut-based structures are cube-shaped with nodes located in the centre of the body (body-centred-cubic (BCC)), and with nodes located in the centre of the cube walls (face-centred-cubic (FCC)), along with their Z variants (BCCZ and FCCZ) containing vertical beams. The above topologies are often chosen due to their simple design, as well as the minimal amount of material needed to manufacture them [12].

Lattice structures can mainly be classified into tensile and bending structures, depending on their internal structure and the number of nodes. This classification can be perform using Maxwell’s stability criterion. Mechanical properties such as energy absorption are much more effective for bending structures. High strength and stiffness are appropriate for tensile structures [13].

Due to the grid shape obtained using lattice structures and the fact that only the axial load is transferred in the element, a fairly high strength/stiffness ratio may be achieved. This dependency means that lattice forms are extremely suitable for projects utilised in aviation, automotive, aerospace and other industries [14].

Complex spatial structures are usually manufactured with 3D printing technology, using methods such as SLS (selective laser sintering), CJP (colour jet printing) or SLA (stereolithography) [15]. SLA technology employs a laser or light source to form photopolymer resins layer by layer to build a lattice architecture. Furthermore, their unit cell structures can be controlled and optimised to achieve the expected mechanical properties suitable for the given application. The use of gradient decomposition in spatial structures significantly modifies their mechanical properties and extends the range of potential applications [16,17,18,19].

Experimental studies [20,21,22,23,24,25,26,27,28,29,30] conducted on printed structures allow for the determination of their specific, and often individual, parameters. Based on the tests, the characteristic mechanical properties of the designed structures, such as fatigue strength [31,32], yield point [33] and critical force [34], are determined. An example of the implementation of such research is presented in [35,36], in which the authors designed and fabricated three different octet-lattice structures of varying densities from polymer resins. The structures were printed using stereolithography (SLA). The mechanical behaviour of these structures under quasi-static and dynamic compressive loads was studied. The mechanical behaviour of the printed octet structures was found to depend on both the relative density and the intrinsic material properties.

Additively manufactured micro-lattice structures offer a unique combination of physical properties such as lightness, high compressive strength, impact resistance and energy absorption. The properties of these structures can be adjusted by changing their shape. Studies [37,38,39] investigated the effect of changing the geometry in a micro-lattice structure on unit energy absorption. Advances in additive manufacturing have simplified the development of spatial structures and made it possible to develop more complex structures. Study [40] classifies different groups of lattices based on their topology and proposes a methodology to improve their energy absorption to weight ratio under compression. Particular attention was paid to the effect of changing the relative density in lattice structures printed with a 3D incremental stereolithography method. The results of the experimental compression test show that a uniform structure with a homogeneous relative density distribution provides the highest initial stiffness of all the printed shapes. However, a graded design with variations in relative density can significantly increase the stiffness and energy absorption capacity of lattices subjected to high compressive stresses. In [41,42,43], the authors also addressed the subject of changing the structure parameters, e.g., geometry, which affected the changes in the energy-intensive characteristics of the analysed structures.

Studies [44,45] showed that the studied additively manufactured structures were able to recover their shape significantly after considerable deformation caused by, for example, an impact. The analysed structures were able to recover their original shape after compression up to 70% of the strain. These results indicate the potential of additive manufacturing as a versatile tool for constructing structures with complex geometries that effectively absorb energy.

The literature review suggests that cellular structures are interesting materials due to the development of additive printing techniques. These techniques enable the development of unique structures varying in material, shape and mechanical properties. The abundance of possibilities for creating such structures requires constant research and analysis by scientists to find the most appropriate design and to achieve the best mechanical properties. The most challenging task for researchers is to develop a range of different lattice structures for use in different areas of life.

Five types of beam-like structures were studied in the research. The structures were printed using stereolithography (SLA). They had regular-cell type topologies with periodically repeating and interconnected elementary cells. The characteristics of the evaluated topologies were as unique and versatile as possible, although maintaining simple designs. There was an example of a topology with quasi-isotropic properties and quasi-irregular structure, as a trabecular topology. The Grid structure, in which beams are oriented only in two perpendicular directions, had an orthotropic characteristic. Other topologies have beams oriented in various directions and under different angles in relation to the compression force applied. Different structures of topologies are subject to different stress conditions, mainly dependent on the beam number and orientation. The hexagonal topology had a similar design to the honeycomb structure, which proved to be very efficient in engineering (e.g., a honeycomb sandwich structures in the aerospace industry) and is inspired by nature.

The aim of the research is to evaluate and compare the compression curves, the history of the compression process and energy-absorption capabilities of the topologies with different characteristics. Based on the obtained data, more efficient structures can be developed in the future research. Moreover, the energy-absorbing data can be utilised for numerical research and energy absorbing structure engineering.

## 2. Description of Sample Preparation

Autodesk Netfabb 2021 software was used to model five different types of 3D-printed lattice structures. The topologies of the structures are shown in Figure 1, and are described as: Grid, Hexagon, Star, Tetra and Trabecular.

Each of them consisted of interconnected beams with circular cross-sections of equal nominal diameters: 1.5 mm. In addition, two flat faceplates, top and bottom, were printed on the faces of the specimens, on opposite sides. For statistical verification, five specimens were tested for each structure. A total of twenty-five specimens were investigated.

The specimen material is Durable Resin V2, characterised by impact and wear resistance and a low friction coefficient. It is recommended for high-impact rigid yet flexible parts, as well as in cases when low-friction surfaces are required. It is especially recommended for prototyping parts that will eventually be made of polypropylene and HDPE (high-density polyethylene). However, it is not recommended for use in environments with temperatures higher than room temperature [46]. The technical parameters for Durable Resin V2 are given in Table 1.

The overall dimensions of the sample were 40 mm × 40 mm × 35 mm, where the first two dimensions are the transverse dimensions and the latter (35 mm) is the height including the faceplates. The geometry of the topology without faceplates is based on a 30 mm × 30 mm × 30 mm cube plane. The maximum shortening of the specimen was set at 15 mm, which corresponds to the engineering strain related to the initial height of the topology, at the level of 50%.

In subsequent stages of the printing process, the components were placed in a bath of isopropyl alcohol for 15 min and then subjected to the curing process, i.e., annealing combined with UV irradiation (Formlabs Inc., Somerville, MA, USA). The process was carried out according to the manufacturer’s recommendations, i.e., at 60 °C for about 60 min. The structures were printed with full raft and supports in the form of beams with a diameter about 0.8 mm. The size of the touch point was 0.6 mm. In the final stage, the support elements were removed manually with precision tools.

The structures were designed in such a manner that the total volumes were as close to each other as possible. It was determined that acceptable sizes should be in the range of 10 cm^3^ to 11 cm^3^. The threshold values were assumed, taking into account the capabilities of the printer and the material, as well as the geometrical parameters of the structures such as basic cell size, number of elementary cells, and diameter of the beams. Detailed parameters of the structure dimensions are compiled in Table 2. The topology volume refers to the volume of the topology material without the faceplates. The total volume refers to the volume of the structure material including the bases. Accordingly, the entire sample and the topology weights were determined using the density parameters provided in Table 1. The cured resin density was 1.13 g/cm^3^. The nominal resolution of the triangular lattice was 0.1 mm for each model geometry of the structure and the radius of the rounded beams was 0.75 mm. In determining the specific absorbed energy (SAE) parameters, the volume/weight of the faceplates was not included because they were not ruptured. The SAE calculation took into account the weight and volume proportional to the deformed part of the topology (about 50% of the initial height of the topology).

A Formlabs Form 2 3D printer (Formlabs Inc., Somerville, MA, USA) employing the SLA method of photo-curing the liquid resin with a 250 mW laser beam was used to produce the structures under study. The thickness of a single layer, applied incrementally to the samples, was 100 µm.

Exemplary photographs of the printed structures are shown in Figure 2. The Grid (Figure 2a) configuration is the most regular structure of the configurations studied. It is based on the structure of ordered elementary cubic cells. The height of one cell is 5.9 mm. The whole structure has highly orthotropic properties, because the beams are aligned in one of two directions: parallel or perpendicular to the load axis.

The Hexagon structure, shown in Figure 2b, is built of hexagonal cells which have mutual sides with the neighbouring cells. The height of one cell is 11.5 mm. Its structure is similar to a honeycomb topology and its beams are positioned at 60° and perpendicular to the load axis.

The third version of the studied configurations is the Star topology (Figure 2c), which resembles a star by its shape of beams intersecting at one point. The beams are arranged spatially in the load axis, and at an angle of 45° to the load axis. Therefore, this is an example of Z-type topology. Ten ends of the beams meet at one point.

The Tetra configuration, illustrated in Figure 2d, is similar in concept to the Star structure, except that it uses more unit beams. Twelve elementary beams meet at one point. The beams are spatially arranged: perpendicular and at 45° to the load axis. There are no vertically aligned beams. The height of one cell is 10.5 mm.

Finally, the Trabecular beam-like configuration was proposed (Figure 2e). The unit elements are arranged in random directions; hence, the topology properties are close to isotropic. The height of the elementary volume is 8.7 mm.

All the topologies studied are characterised by a non-uniform cross-sectional area. Figure 3 illustrates the variation of the changes along the vertical coordinate. The zero value of this coordinate was set at the mid-height of the topology; therefore, the abscissa values in graph in Figure 3 vary by ±15 mm. Cross-sectional measurements were taken at 0.1 mm intervals. The study did not take into account the cross-sectional area of the faceplates. Their dimensions are nominally 40 mm × 40 mm for all the samples. The area of the plates is 1600 mm^2^ and their thickness is approximately 2.5 mm.

A detailed analysis of the minimum, average and maximum cross-sectional area values for all topologies is given in the form of a bar chart in Figure 4. The Grid structure proved to assume the smallest, and at the same time, the largest value of the minimum cross-sectional area, which exhibits the greatest variability in this respect. The structure is built of beams aligned perpendicular to the load axis, which are arranged in a cube edge plane.

The Hexagon, Star and Tetra topologies are characterised by a similar maximum cross-sectional area value of approximately 250 mm^2^. These three structures performed similarly in this study. The beams are arranged in them in various ways. However, the Star topology has a significantly smaller minimum cross-sectional area compared to these three structures, which results from the crossing points of all the beams. On the other hand, the Tetra structure, which contains beams positioned perpendicular to the load, has the largest cross-sectional area in this group.

When comparing the mean cross-sectional areas, it can be noticed that they are similar for the compared topologies. The cross-sectional area averaged for all topologies is about 140 mm^2^. It is worth noting that in the Grid topology, this value is lower than for the others and is around 124 mm^2^. The Trabecular structure shows the least variability, and the average cross-section is close to the overall average. For determination of the engineering stress on the compression curves, it was decided to consider the minimum cross-sectional area of each topology (Figure 4).

Verification of geometric accuracy involved random measurement of the diameters of 10 beams on each wall of the structure using a digital calliper. A total of 40 measurements were performed for three samples of a specific structure type.

For each structure studied, the average value of the measured quantities, marked with a symbol ${\bar{d}}_{0}$, was determined. Based on the results obtained, the global relative error for the printed spatial structures was determined $\delta_{gi}$.
(1)$$\delta_{gi}=\frac{\left|d-\left({\bar{d}}_{01}+{\bar{d}}_{02}+{\bar{d}}_{03}\right)/3\right|}{d}\times 100\%$$
where *d*—the exact beam diameter (*d* = 1.5 mm), ${\bar{d}}_{01}$ is the average of the measurement for the first sample, ${\bar{d}}_{02}\;$is the average of the measurements for the second sample, and ${\bar{d}}_{03}\;$ is the average of the measurements for the third sample.

The data presented in Table 3 demonstrate that the highest print error was recorded for the Trabecular structure (12.7%) and the lowest for the Grid structure (4.7%). The relative error, averaged over all spatial structures, was approximately 8.7%. The magnitude of the error may be related to the complexity and sophistication of the geometry, the type of material from which the prints were made and the post-printing process. For the Grid structure, the print geometry is less complex compared to the Trabecular topology.

The averages of the measured diameters of the beams, of which all structures are composed, are of slightly underestimated values, ranging from 1.31 to 1.43 mm, compared with the nominal value of 1.5 mm. This detail is of great importance, especially when developing and analysing numerical models. The inclusion of realistic geometric quantities can positively affect the correlation of experimental and numerical results.

## 3. Test Methodology

Experimental studies reported in the literature mainly take place under quasi-static loading conditions, with few examples of studies under dynamic loading conditions. The most common tests performed on spatial structures are uniaxial compression tests. A similar measurement scheme was adopted in this study. A Zwick/Roell testing machine (ZwickRoell GmbH & Co. KG August-Nagel-Straße, Ulm, Germany) with a KAPPA 50 DS loading system was used for the tests. The samples were compressed at a constant loading rate, which was set at 5 mm/min. The maximum shortening of the sample was set at 15 mm, which is approximately 50% of the height of its topology. During the tests, the force was recorded using an integrated load cell and the displacement of the grips (sample shortening) was recorded at a frequency of 10 Hz. A black and white camera integrated into the machine was used to record the image sequence. A picture of the testing machine with a specimen placed in it is shown in Figure 5. The load was applied using flat machine supports, perpendicular to the planes of the faceplates installed on the opposite sides of the sample. One of the flat supports of the machine used was equipped with a hinge, which reduced surface unevenness.

## 4. Analysis of Test Results and Graphs of Structure Elongation Curves

A comparison of the compression process of the studied topologies was performed. The results of the calculations are given in Table 4. As a result of statistical analysis of the data of the five samples tested, the upper and lower bounds of the sets were calculated for 95% confidence intervals (assuming a Student’s *t*-distribution). In a further step, the two extreme graphs were rejected. Based on three representative samples, new mean values and their standard deviations were determined. The mean value and initial maximum stress, as well as the corresponding stress values for 25% and 35% shortening are given.

To determine the engineering stress of the compression curves, the minimum cross-section area of the given topology was inserted based on the data presented in Figure 4. The engineering strain was referenced to an initial nominal topology height of 30 mm. The maximum shortening of the specimen was set at 15 mm, which corresponds to the engineering strain of 50%.

The absorbed energy is the area under the graph of the force–displacement curve. The specific absorbed energy was calculated by dividing AE by the mass of the deformed part of the topology, excluding the weight of the face plates. The efficiency of the crushing force is the quotient of the initial force extreme to the average force value.

Figure 6, Figure 7, Figure 8, Figure 9 and Figure 10 show the compression curves in the engineering stress versus engineering strain coordinates of three representative samples for all topologies tested. They present sequences of photographs for the selected compression phases, which are assigned to corresponding areas of the curve marked with a red rectangular envelope and an arrow. The camera image was synchronised with the course of the test.

Grid sample compression curves (Figure 6) start with the linear part. As shown in the photographs, during the initial loading phase of the Grid structure, there was uniform strain of the entire sample. The maximum initial stress is, according to statistical analysis, in the range of approximately 10.5–18.5 MPa (force of 450–800 N). The load extreme occurs for a shortening of approximately 1.5 mm. The deformation process is dominated by compression of the vertical elements combined with tension of the horizontal beams. When the initial maximum is exceeded, the force gradually decreases as the vertical beams of the structure buckle and lose their stability. When the minimum load force is reached, with a shortening of 5 mm, there is a gradual increase in the load force, caused by an increase in the packing of the material in the increasingly smaller volume of the structure contour. The photographs in Figure 6 prove that cell rows in the structure worked one after the other, but not consecutively. The cell rows at the upper and lower supports were damaged first. The row of central cells only became visibly deformed at the final stage of compression.

The sample graphs of the Hexagon configuration, presented in Figure 7, show a smooth course. As the photographs show, at the beginning of the test, cell deformation occurred evenly throughout the structure. The side walls of the topology, which are more numerous, are angled at 60° and work in bending combined with compression while gradually losing stability. There are no beams positioned in the load axis here. This structure implies that there is no initial load extreme. The noticeable increase in force, after shortening of about 9 mm, is attributed to the strong shortening of the structure and the progressive packing of the material in an increasingly smaller volume. It can be observed in the photographs that the hexagonal cells have flattened. They have shrunk in height in the load axis and increased in width. In the first instance, the layers at the lower support were deformed the most. The extreme walls of the lower layer no longer engage with any of the walls of other cells and have acted as initiators of destruction. In the final compression phase, it can be observed that the sample contour has expanded uniformly along the height; however, this is slightly less at the supports. The maximum stress values, around 6 MPa on average, were recorded at the end of the curve (Figure 7).

The Star structure showed repeatable results during the tests. The curves are smooth, although not as much as in the case of the Hexagon topology. In the Star structure, some of the beams are axially aligned to the load direction and are subjected to buckling. Thus, the graph clearly shows an initial load extreme (mean 459 N/7.5 MPa), which occurs for a displacement of approximately 2.5 mm. However, it is not as sharp as for the Grid structure, in which a significant proportion of the vertical elements work in compression and buckling. According to the illustrations shown in Figure 8, this topology undergoes a kind of barrel effect. Bars aligned parallel to the load axis are axially connected to each other and form a single bar with a length equal to the height of the topology. Beams positioned on the sample edges are subjected to buckling.

In general, for the classic buckling of a beam, the largest deflection occurs at mid-height. Accordingly, the most susceptible area appears to be the middle layers of the structure. Most transversely positioned elements work in bending. After the initial load extreme, the stress decreases to about 6 MPa, which corresponds to a sample shortening to about 7 mm. The maximum force values were obtained at the end of the test, with a high packing of material, and were in the range of 500–560 N. The last of the photograph sequence shows that the sample expanded significantly at the lower support.

The Tetra configuration has horizontal beams, positioned exactly perpendicular to the load axis, which are mainly in tension during compression. The first characteristic point of the curve (the initial extreme of the force) can be observed when a displacement of 3 mm (about 10% of the strain) is reached (Figure 9).

When the linear range is exceeded, there is a decrease in strength, typically as in case of other structures. However, this structure is characterised by force oscillations in the middle of the compression curve. The laterally aligned beams, mainly from the middle layer, buckle, causing a loss of stability and collapse while working in bending and compression. This results in an approximately linear decrease in force until a shortening of approximately 8 mm (26% of the strain) is achieved. Subsequently, the neighbouring layers (from above and below) with the already strongly deformed middle layer undergo major deformations. In the middle layer, there is a progressively more considerable compression of the material, resulting in a further increase in force. There is a second local extreme here, which can be observed with shortening in the range of 9 to 12 mm depending on the sample (approximately 33% of the strain). In the subsequent compression phase, there is a loss of stability of the beams from these layers, which manifests in a further decrease in load. The maximum values were obtained in the final stage of the test and range from 260 N to 370 N (below 4 MPa). Notably, the edge layers at the upper and lower supports have not been significantly deformed. The most compressed layers in the middle eventually expanded slightly in the direction transverse to the load.

When analysing the deformation course of the Trabecular structure from the photographs in Figure 10, it should be noted that it is characterised by the greatest degree of irregularity and is a stochastic structure with near-isotropic properties. In compression, this topology works globally and deforms uniformly. There is no distinctive area here. Both at the supports and at the centre of the sample, the structure works in a similar way. The photographs show that its outer contour undergoes an even transverse expansion.

The compression curves of the Trabecular configuration are characterised by a continuous smooth course of forces, with a typical distinguished initial linear part, as in the case of all other topologies. For a strain of approximately 9% (shortening of less than 3 mm), there is a slight decrease which continues until a strain of 18% (approx. 5.5 mm) is reached. Further loading of the structure results in a gradual increase in load, due to compression of the material and the closure of voids in compression. The maximum force generated was recorded at the end of the test; based on statistical tests, this ranged from 510 N to 707 N (9.4 to 6.8 MPa, respectively).

## 5. Comparison of Test Results

A graphical comparison of the energy-absorbing parameters characterising studied topologies is presented in Figure 11. It should be noted that the studies presented in this paper are comparative, and although the target topologies were most likely built of other materials, certain conclusions can also be used for the target structures.

The Grid structure is characterised by a significant initial extremum, which is manifested in the calculations by a high crushing force effectiveness factor of 1.77. This value should ideally be as close to uniformity as possible. This is approximately the case for the Hexagon topology (1.10; see Figure 11), for which the load increase is smooth. The Trabecular topology is described by a factor of less than one (0.94 precisely), due to a fast final increase in force resulting from high compression of the material. From the graph in Figure 10, it can be seen that the useful shortening (measured up to the moment of force increase) of the Trabecular structure is low, around 26%. Therefore, this topology reached the limit of its usefulness during compression. For other topologies, the point of sharp increase in force is not visible, which means that it was not reached and the useful shortening in this case is higher than 50%. Accurate determination of this value requires increasing the levels of sample shortening during testing.

Energy-absorption parameters are important from the point of view of the potential application of the 3D structures studied. The most important of these are absorbed energy (AE) and specific absorbed energy (SAE), which can typically be associated with the weight or volume of the destroyed portion of the sample. The absorbed energy is calculated as the area under the force–displacement graph. Analysis of the bar graph shown in Figure 11 shows that the Tetra topology achieved the lowest AE, and consequently, SAE in the study, equal to 2.82 J and 1.21 J/g, respectively. The average force recorded for this topology was low and equalled 195.5 N (2.15 MPa), and the graph is non-uniform (approximately sinusoidal). The Star and Hexagon topologies achieved the highest AEs with 6.14 and 5.92 J, respectively. The Trabecular topology also achieved a relatively high AE of 5.4 J (Figure 11). The Grid topology reached a low curve after a large initial increase in force. AE was equal to 4.85 J. Both the volumes and weights of the tested structures are similar (Table 2), which was a condition for allowing the prints to be compared with each other; therefore, the AE and SAE ratios are similar.

Summary graphs of compression force versus displacement, and stress–strain are presented in Figure 12 and Figure 13, respectively. Comparison of these two summary graphs shows that the relationships between the curves of the examined topologies change, particularly in the case of the Grid topology, characterised by a significantly lower minimum cross-sectional area than the others (Figure 4). This results in the increased stress relative to other structures. However, the Star topology presents the largest minimum cross-sectional area and, in this case, the stress curve is relatively lowered. However, the nature and shape of the graphs in Figure 12 and Figure 13 remain identical.

Grid and Star structures show the highest slopes of the linear range and the largest values of initial force extremes in the stress–strain graphs. Both of these structures contain beams arranged in the load axis, the shortening of which requires the application of a higher load. When stability is lost, there is a significant decrease in force, due to buckling of the beams. The Grid structure contains more beams positioned in this way; therefore, in this case, the initial topology stiffness measured by the slope of the linear stage of the curve is the greatest. Moreover, the maximum initial force is higher, and its reduction is steeper.

Both Hexagon and Tetra topologies do not contain beams oriented in the load axis. Therefore, their transition from the linear part is smooth. Comparison of the structures of these two topologies shows that the angle of deflection of the beams from the load axis for the Tetra structure is greater, which causes bending in the dominant load. This results in a low value of the loading force in the middle part of the graph. However, the contribution of compression in the stress state of the Hexagon topology is higher. Hence, the loading forces are higher. The Tetra topology is the most susceptible structure of the systems studied. It also has the lowest initial stiffness of the linear range.

The Hexagon structure presents a considerably smooth course. The flat plateau in the central part of the graph is clearly outlined.

The nature of the performance of the Hexagon and Trabecular structures is similar in the linear range (slope of the section in the stress–strain graphs). Their stiffness is relatively low. The Trabecular structure exhibits quasi-isotropic properties. The curve in the central part is declining. In this structure, the compression of the material causes a rapid increase in load at the end of the curve, the fastest of the whole set.

Based on the data summarised from the measurements, it can be concluded that the Grid and Hexagon structures provided the least convergent results compared with the other structures, confirmed by the standard deviation values displayed in Table 4.

Discrepancies in the obtained results may arise from printing inaccuracies. When printing such complex structures, the global relative error calculated from the beam measurements was approximately 8.7% (Table 3). Taking into consideration the individual structures, this error ranged from 4.7% (Hexagon) to 12.7% (Trabecular). Thus, the print error for Grid and Hexagon was the lowest (4.7% and 8.0%, respectively) and the scatter of results was the largest. This contradiction is convincingly explained by the presence of vertical beams which, due to the presence of buckling in compression, can cause these topologies considerably sensitive to the variations of the beams dimensions resulting from the precision of their printing.

On the other hand, the quasi-isotropic Trabecular topology shows the highest print error rate of 12.7% and much better repeatability of tensile curve results (Figure 10). This structure is chaotic, and due to its character, is less susceptible to print quality.

The Star structure is also characterised by the good repeatability of tensile curve results. The measurement error is also significantly low (6%). Bending accounts for a large proportion of the stress state in these topologies. It can be concluded that this loading state is less sensitive to the quality of print.

In the final stage of structure compression, a gradual increase in force can be observed, which is partly related to the strong strain of the elements as well as to the mutual contact of the internal elements of the structures, their compression and increased packing. Due to shortening with constant speed, the volume decreases, whereas the amount of material in the structure is constant.

The highest end-load values were recorded for Trabecular, Star and Hexagon samples and are 600 N, 532 N and 493 N, respectively (mean of three representative samples). Grid and Tetra samples have average end-load values of 357 N and 310 N, respectively. The highest increases in force after central plateau were recorded for Trabecular, Star and Tetra topologies.

None of the samples tested cracked. During compression of the topologies studied, significant deformations occurred due to high specimen shortening. The elongation of Durable Resin V2 is 55%, which proved to be sufficient to study the structures described. This material is a good choice for prototyping such structures.

## 6. Summary and Conclusions

In the presented study, the SLA-type additive printing technique was used to design and print five diverse and unique 3D topologies from Durable Resin V2. The structures were subjected to a quasi-static axial compression test. Their basic material and energy-absorbing properties were determined from the compression curves recorded during the tests. The deformation processes of the beams and the results obtained from the tests were compared. The conclusions for each structure are as follows:The print quality of individual structures is highly dependent on the topology of the structure in question and the orientation in the printer workspace. The highest print error was recorded for the Trabecular structure (12.7%) and the lowest for the Grid structure (4.7%). The scatter of results depends not only on the quality of the print, but primarily on the loading state. Topologies involving vertically oriented compression and buckling beams presented the greatest scatter in the results. The Star and Tetra topologies, which show a large share of bending, are less sensitive to print quality;Grid and Star topologies show the largest values of the initial local force extremes and the highest stiffness of the linear range. Both of these structures contain beams aligned with the load axis, the shortening of which requires the application of a higher load. After the loss of stability, due to buckling of the beams, a decrease in force occurs. In the case of the Grid topology, the maximum force is the highest and its reduction is steep;Both Hexagon and Tetra topologies do not contain beams oriented in the load axis. Their transition from the linear part is considerably smooth. The contribution of compression in the loading of the Hexagon topology is higher; hence, the loading forces are also higher;The Tetra topology is the most susceptible structure of the systems studied. This configuration contains horizontal beams, positioned accurately perpendicular to the load axis, which are mainly in tension during compression. This structure is characterised by force oscillations in the middle of the compression curve. The laterally aligned beams, working in bending and compression, are subjected to buckling, which causes a loss of stability and collapse. Bending is the dominant load in this case, which results in a low value of the loading force in the middle part of the graph;It should be noted that the Trabecular structure is characterised by the greatest degree of irregularity and is a stochastic structure with near-isotropic properties. In compression, this structure works globally and deforms uniformly. The level of crushing force in the middle area is higher than for the Grid structure;After analysing the energy-absorbing parameters, it can be concluded that the Star and Hexagon topologies achieved the highest AE, with 6.14 and 5.92 J, respectively. The Trabecular topology also achieved a relatively high AE. The lowest AE of 2.82 J was recorded for Tetra topology. Both the volumes and weights of the topologies tested are similar, which was a condition for allowing a mutual comparison; therefore, the AE and SAE ratios are similar;The Grid structure is characterised by a high crushing force effectiveness factor of 1.77, due to a significant initial force extremum. This value should ideally be as close as possible to unity. This is approximately the case for the Hexagon and the other topologies;The useful shortening of the Trabecular topology, measured up to the moment of the rapid crushing force increase, is considerably low, equal to around 26%. Therefore, this topology reached the limit of its usefulness during compression. For other topologies, the point of sharp increase in force was barely reached, which means that the useful shortening in this case is higher than 50%.

## Figures and Tables

**Figure 1 materials-15-00378-f001:**
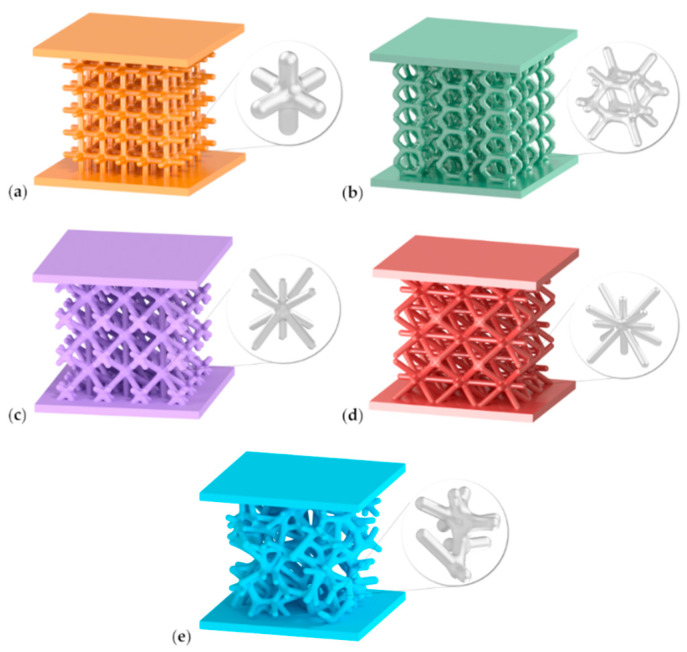
Lattice structures designed in Autodesk Netfabb 2021, with the following topology: (**a**) Grid; (**b**) Hexagon; (**c**) Star; (**d**) Tetra; (**e**) Trabecular, together with enlarged beam-like shapes.

**Figure 2 materials-15-00378-f002:**
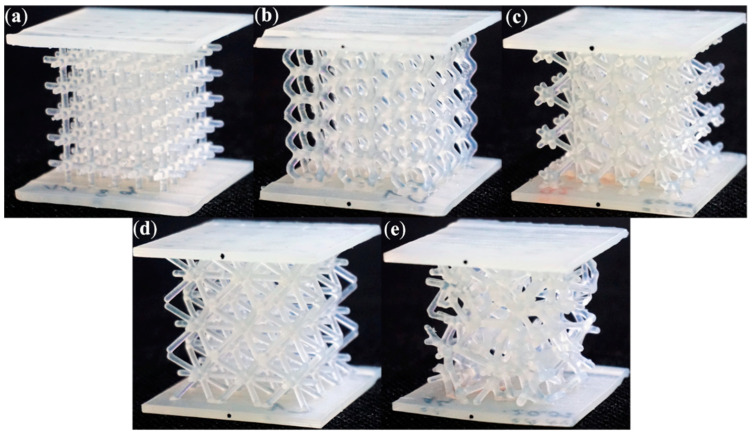
Photographs of the printed structures: (**a**) Grid; (**b**) Hexagon; (**c**) Star; (**d**) Tetra; (**e**) Trabecular.

**Figure 3 materials-15-00378-f003:**
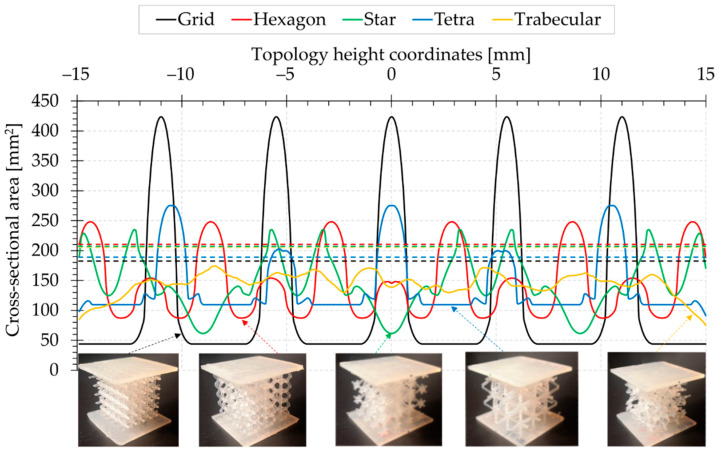
Cross-sectional area variation of the topology along the height, excluding faceplates. The dashed lines indicate average values.

**Figure 4 materials-15-00378-f004:**
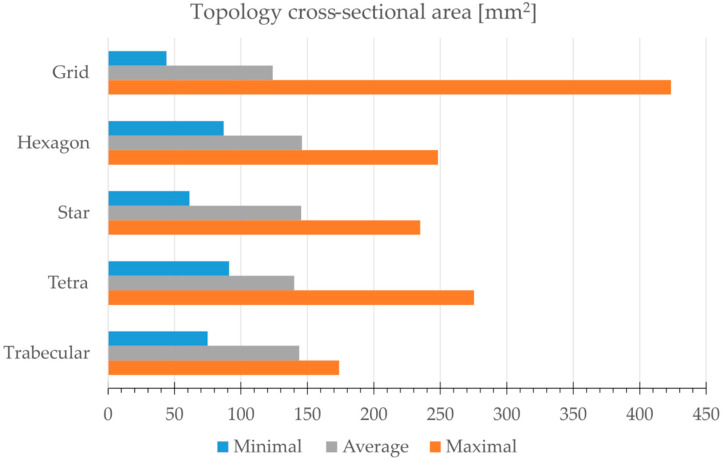
Graphical comparison of variation and mean cross-sectional area of all topologies.

**Figure 5 materials-15-00378-f005:**
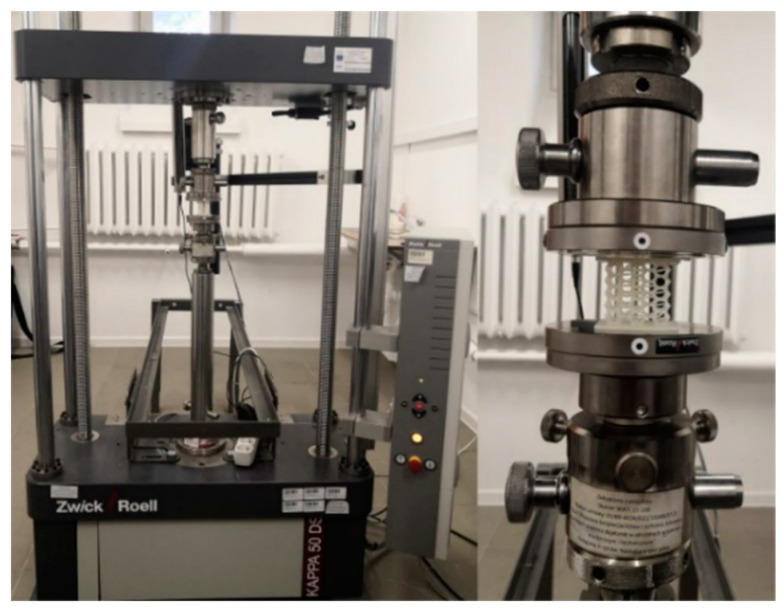
The testing machine with a specimen placed in it.

**Figure 6 materials-15-00378-f006:**
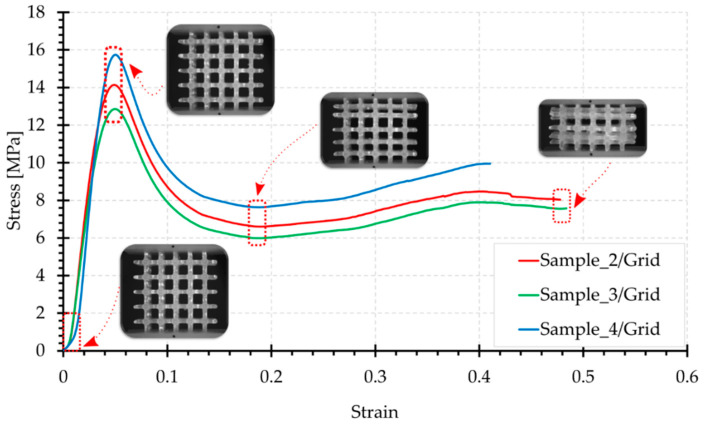
Compression curve for the Grid structure with photographs of the topology at selected test phases.

**Figure 7 materials-15-00378-f007:**
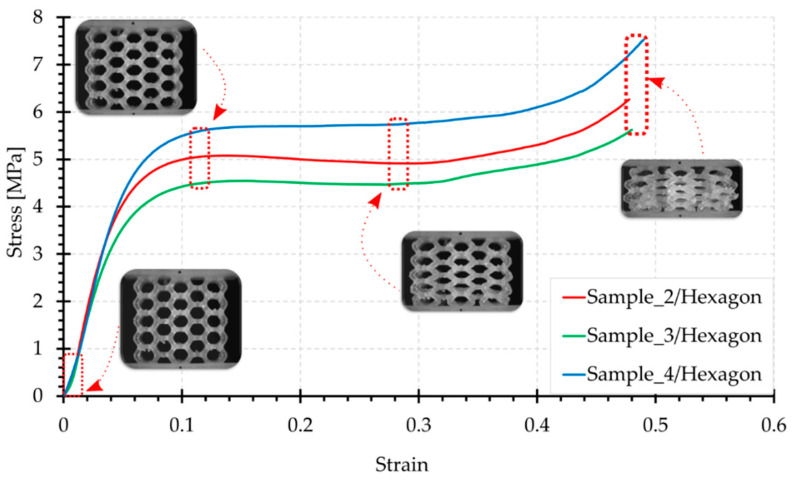
Compression curve for the Hexagon structure with photographs of the topology at selected test phases.

**Figure 8 materials-15-00378-f008:**
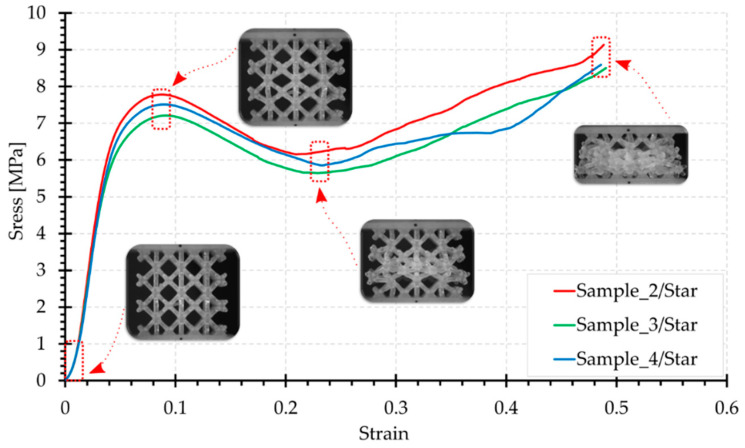
Compression curve for the Star structure with photographs of the topology at selected test phases.

**Figure 9 materials-15-00378-f009:**
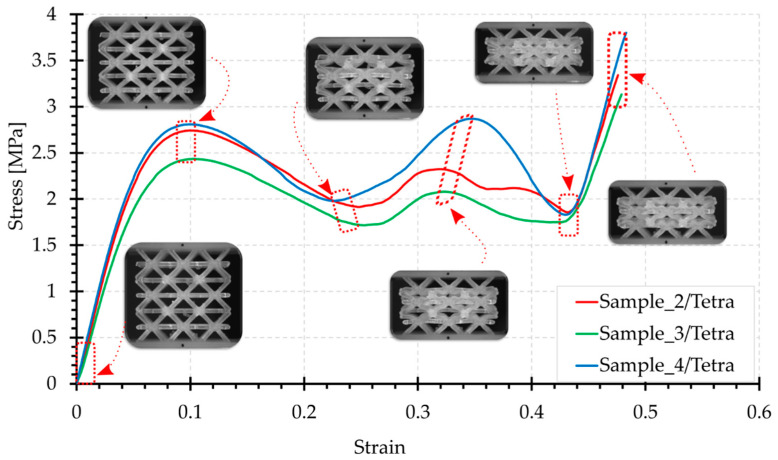
Compression curve for the Tetra structure with photographs of the topology at selected test phases.

**Figure 10 materials-15-00378-f010:**
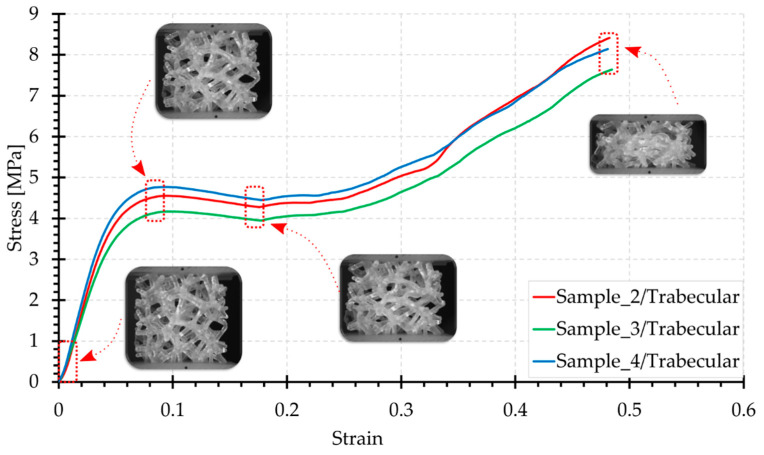
Compression curve for the Trabecular structure with photographs of the topology at selected test phases.

**Figure 11 materials-15-00378-f011:**
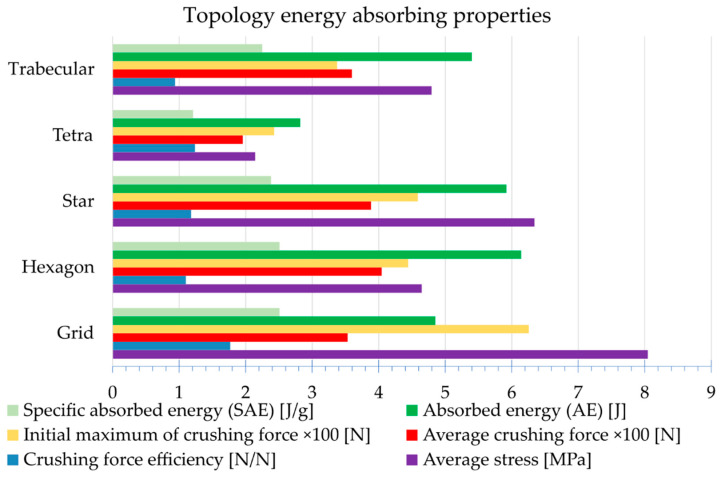
Graphical comparison of the energy absorbing parameters of the topologies studied.

**Figure 12 materials-15-00378-f012:**
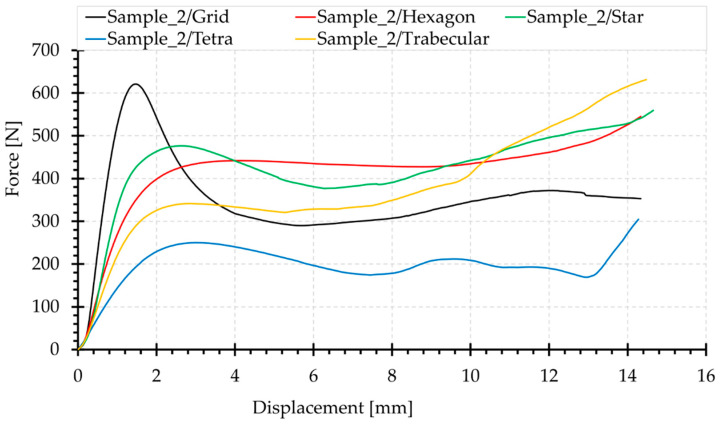
Summary of representative compression force–displacement curves, for the topologies analysed.

**Figure 13 materials-15-00378-f013:**
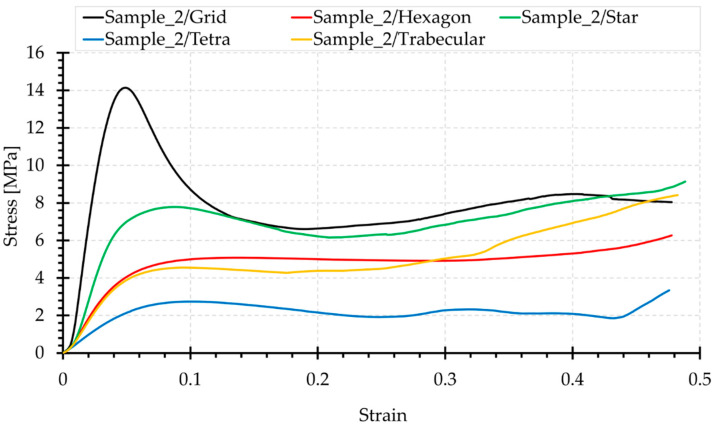
Summary of representative compression sigma–epsilon curves, for the topologies analysed.

**Table 1 materials-15-00378-t001:** Technical parameters for Durable Resin V2 [46].

Properties	Values
Tensile strength	28 MPa
Tensile modulus	1.0 GPa
Elongation	55%
Heat deflection temperature, HDT	41 °C
Liquid density	1.06 g/cm^3^
Cured density	1.13 g/cm^3^

**Table 2 materials-15-00378-t002:** Comparison of structures with print parameters and nominal dimensions.

Topology	Beam Diameter [mm]	Volume of Topology [cm^3^]	Total Volume [cm^3^]	Weight of Topology [g]	Total Weight [g]
Grid	1.5	3.76	10.11	4.24	11.42
Hexagon	1.5	4.49	10.78	5.07	12.18
Star	1.5	4.51	10.77	5.10	12.17
Tetra	1.5	4.29	10.60	4.84	11.98
Trabecular	1.5	4.39	10.72	4.96	12.11

**Table 3 materials-15-00378-t003:** Results of statistical analysis of beam diameter measurements of the tested topologies.

**Structure Topology**	${\bar{d}}_{01}\;\left[\mathrm{mm}\right]$	${\bar{d}}_{02}\;\left[\mathrm{mm}\right]$	${\bar{d}}_{03}\;\left[\mathrm{mm}\right]$	$\delta_{gi}\;\left[\%\right]$
Grid	1.41	1.49	1.40	4.7
Hexagon	1.40	1.38	1.35	8.0
Star	1.40	1.41	1.43	6.0
Tetra	1.31	1.34	1.31	12.0
Trabecular	1.31	1.32	1.39	12.7
Global average:				8.68

**Table 4 materials-15-00378-t004:** Statistical calculations of the analysed parameters for each topology.

Topology	Description	Initial Maximum Force [N]	Displacement for Initial Maximum Force [mm]	Average Force [N]	Maximum Initial Stress of Minimum Section [MPa]	Average Minimum Cross-Sectional Stress [MPa]	Minimum Cross-Sectional Stress for 25% Shortening [MPa]	Minimum Cross-Sectional Stress for 35% Shortening [MPa]
Grid	Average *	625.68	1.49	353.34	14.25	8.04	7.06	8.27
	Standard deviation *	63.46	0.01	40.59	1.44	0.92	0.84	0.92
	Upper bound of the set **	801.49	1.51	484.75	18.25	11.04	9.88	12.03
	Lower bound of the set **	456.89	1.45	241.74	10.40	5.50	4.46	4.96
Hexagon	Average *	444.49	4.64	404.40	5.11	4.65	5.05	5.21
	Standard deviation *	50.16	0.52	45.12	0.58	0.52	0.63	0.61
	Upper bound of the set **	532.76	6.01	495.79	6.12	5.70	6.10	6.64
	Lower bound of the set **	342.60	3.48	318.27	3.94	3.66	3.92	4.01
Star	Average *	458.86	2.60	388.40	7.49	6.34	5.99	6.95
	Standard deviation *	16.73	0.18	18.77	0.27	0.31	0.31	0.40
	Upper bound of the set **	533.05	2.75	446.16	8.71	7.29	6.91	8.18
	Lower bound of the set **	399.46	2.40	343.04	6.52	5.60	5.27	5.95
Tetra	Average *	242.69	3.03	195.54	2.66	2.15	1.90	2.34
	Standard deviation *	18.22	0.04	18.31	0.20	0.20	0.17	0.47
	Upper bound of the set **	294.20	3.32	231.71	3.23	2.54	2.22	2.84
	Lower bound of the set **	188.73	2.93	154.09	2.07	1.69	1.51	1.72
Trabecular	Average *	337.38	2.86	359.77	4.50	4.80	4.45	5.78
	Standard deviation *	22.81	0.04	18.37	0.30	0.24	0.26	0.36
	Upper bound of the set **	396.78	3.09	414.41	5.29	5.52	5.18	6.70
	Lower bound of the set **	271.83	2.74	307.92	3.62	4.10	3.69	4.71

* values calculated for three representative topology samples, ** values calculated for five topology samples tested assuming Student’s *t*-distribution.

## Data Availability

The data presented in this study are available on request from the corresponding author.

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
