# Peer review of "Experimental Research of Selected Lattice Structures Developed with 3D Printing Technology"

_materials, 2022, doi:10.3390/ma15010378_

Round 1
Reviewer 1 Report
1) The manuscript looks like a test report rather than an academic paper. I could not see what problems authors tried to solve from the Abstract part, the Introduction part, or/and the Conclusion part of the manuscript. Authors only tried to tell the audience what they had done.
2) Did authors read their manuscript carefully before submitting it? "Error! Reference source not found" were displayed in the manuscript for several times. Some figure numbers were marked wrongly.
3) Some Tables and Figures described the same experimental results. Please keep one type only.
4) What goal is your experiment? Please make it clear in the manuscript.
5) Did the five structures in the manuscript need supports during 3D printing? Could these supports be removed easily and completely if they existed?
5) The Conclusion part of the manuscript is too long...
Author Response
Thank you very much for all the comments submitted. They are very valuable. By following the suggestions, the authors improved the overall quality of the manuscript. Please see the attachment.

Reviewer 2 Report
The paper regards a comparative study of 3D printed resin lattice structures. The overall volume of the tested typologies of structures is kept constant in the comparison and the energy absorption capabilities are measured.
Overall, the manuscript is clear and the results of the experimental tests are properly evidenced. The conclusions of the work are well reported.
Some issues arise in the definition of the experimental approach and, consequently, the following aspects should be considered in the revised version of the manuscript:
- Section 2, explain the reasons for the choice of those topologies of structures.
- Page 3 line 106, is the cross-section of the beams circular? In that case, the authors should refer to the diameter instead of the thickness.
- Page 4, “It was determined that acceptable sizes should be in the range 133 of 10 cm3 to 11 cm3”. How did the authors determine this threshold value?
- Table 4, why the measured values of diameter are all below the target value of 1.5 mm? Explain or try to justify this evidence.
- Page 8 line 239, better explain the measure of the specimen deformation. Was a system of strain gauges employed? How was it positioned on the specimen? Was any problem of planarity of the specimen, i.e. load misalignment, detected?
- Section 4, it would also be interesting to evaluate and compare the stiffness (i.e. the slope of the initial linear stage of the stress-strain curve) of the different typologies of specimens.
- Some references to figures and tables in the manuscript do not work and others are wrong.
- Check the text to correct typos and misspellings.
- The scientific literature about the lattice structures is considerably wide and other references should be added: https://doi.org/10.1016/j.euromechsol.2021.104291; https://doi.org/10.1016/j.compstruct.2020.111985
Author Response

(The authors gave the same response as above.)

Round 2
Reviewer 1 Report
The revised manuscript looks fine.
Reviewer 2 Report
The raised points have been discussed and solved.
As far as I am concerned, the paper can be accepted for publishing.